# Nutrient Content of Micro/Baby-Green and Field-Grown Mature Foliage of Tropical Spinach (*Amaranthus* sp.) and Roselle (*Hibiscus sabdariffa* L.)

**DOI:** 10.3390/foods10112546

**Published:** 2021-10-22

**Authors:** Albert Ayeni

**Affiliations:** Plant Biology Department, School of Environmental and Biological Sciences, Rutgers University, 59 Dudley Road, New Brunswick, NJ 08901, USA; aayeni@scarletmail.rutgers.edu; Tel.: +1-856-279-8641

**Keywords:** microgreens, baby greens, mature vegetables, nutritional elements, macronutrients, micronutrients

## Abstract

Micro/baby-greens are gaining popularity in human diets as functional foods that deliver superior nutritional values and health benefits to consumers. This study conducted multiple times between 2017 and 2019 under greenhouse conditions and in the field at Rutgers University, New Brunswick, New Jersey, USA, showed that micro/baby-greens from tropical spinach (*Amaranthus* sp.) and roselle (*Hibiscus sabdariffa* L.) are rich in digestible carbohydrates, digestible protein, and dietary fiber. On dry weight basis, both vegetables have high relative percentages of P, K, and Mg; and relatively high ppm Fe, Mn, and Zn. Foliage tissues of both species are relatively low in total fat, Ca, and Cu. Between 10 and 20 days after sowing (DAS), percent digestible carbohydrates in fresh foliar tissue increased 100% in tropical spinach and 50% in roselle, while digestible protein dropped 21% in tropical spinach and 50% in roselle. Compared to field grown mature foliage, greenhouse-grown micro/baby-greens were lower in digestible carbohydrates and Ca but higher in digestible protein, P, K, Mg, Fe, Mn, and Zn.

## 1. Introduction

Microgreens, also called “vegetable confetti” [1], belong to the group of plant foods classified as “functional foods” because they possess particular health promoting or disease preventing properties that are additional to their normal nutritional value [2] Morphologically, microgreens are plant seedlings that are between the cotyledonary and the first fully formed primary (true) leaf stages of growth [1]. Commercially and in culinary terms, they fall between the “sprout” and “baby green” vegetable packaging [3]. For plants with tiny seedlings, such as tropical spinach (*Amaranthus* spp.), the microgreen growth stage may be stretched a little longer than the first true leaf stage. While sprouts need water, air, and food reserves in the seed cotyledons (for dicot plants) or endosperm (for monocots) to germinate, emerge from the seed (or grain), and grow; in addition to these elements, microgreens need light for photosynthesis and nutrients from the growth media. Sprouts may be cultured exclusively in moist soilless media with or without light. However, microgreens rely on the food reserves in the sprout as well as nutrients in the growth media and light for photosynthesis to support the initial stages of metabolism needed to provide energy for subsequent vigorous growth of the microgreen, baby green, and the mature plant. The microgreens quite closely reflect the nutrient density of the sprout from which they are derived.

Microgreens are gradually gaining commercial attention globally as nutrient dense seedlings capable of supplying high nutritional and health values at relatively small consumption quantities compared to mature vegetables [3,4,5,6,7,8,9,10,11,12]. In the United States, interest in microgreens has increased 100% since 2004 [7] with Montana (#1–100%), Hawaii (#2–92%), and Vermont (#3–75%) topping the list of states with high interest in microgreens. On the same scale New Jersey (NJ) ranks #45 (of 50) with relatively low interest (32%) in microgreen production. It is desirable to add these nutrient dense plant sources to our food basket in NJ and elsewhere around the United States to promote the vitality of our workforce and enhance our agricultural economy.

The Ethnic Crops Research Program at Rutgers University is interested in documenting the nutritional value of top ethnic crops in its collection starting with tropical spinach (*Amaranthus* spp.) and roselle (*Hibiscus sabdariffa* L.) as micro/baby-greens and mature field grown vegetables. The objective was to determine the nutritional quality of micro/baby greens of tropical spinach and roselle compared with the mature field-grown plants. We compared the nutritional content of these crops, including digestible carbohydrates, digestible protein, total fat, dietary fiber, and elemental macro-and micro-nutrients as both greenhouse-cultured micro/baby-green and as mature field grown vegetables. In the greenhouse, microgreens were extended to the early baby-green stage to be able to track nutritional trends over 20 DAS, while the young seedling remained succulent. For this reason, the nomenclature “micro/baby-green” was used to describe the greenhouse-raised plant seedlings in this study.

## 2. Materials and Methods

### 2.1. Micro/Baby-Greens

Between 2017 and 2019, multiple 10- to 20-day cycles of tropical spinach and roselle micro/baby-greens were raised in the greenhouse at the New Jersey Agricultural Experiment Station (NJAES) greenhouses on Cook Campus, Rutgers University, New Brunswick, NJ. The computer regulated growth conditions were 75–85 °F temperature, 14-h high-pressure sodium (HPS) light/day, and 75–85% relative humidity. The HPS light supplemented the sun and natural light that transmitted through the glasshouse. Tropical spinach (Caribbean Red selection) seeds were obtained from previously processed seeds harvested from ongoing ethnic crop research plots at Rutgers’ Horticulture Farm 3, East Brunswick, NJ. Roselle seeds (Indian Red-Red cultivar) were purchased from Seeds of India (Marlboro, NJ, USA). For tropical spinach, micro/baby-greens were raised in black 1020 trays (Greenhouse Megastore, Danville, IL, USA) half-filled with Pro-Mix (Premier Tech Horticulture, Quakertown, PA, USA) potting mix. In each of nine trays, approximately 3 g of seed was carefully spread by hand evenly across the surface of the potting mix and worked lightly into the mix. For roselle, nine 48-cell insert trays (Greenhouse Megastore, Danville, IL, USA) were used to raise micro/baby-greens. Trays were filled with potting mix, then two roselle seeds were placed about 1.5 cm deep in each cell using the blunt end of a ballpoint pen; and covered. For both tropical spinach and roselle, trays were watered gently with a sprinkler until water started to drip at the bottom of the flat. Seeded trays were set on greenhouse bench and watered once in two days for the first 10 DAS, then increased to once a day until study was terminated 20 DAS. After emergence trays were watered once (about 6–7 DAS) with 3 g/L (0.4 oz/gal) 20-20-20 NPK solution. For each type of micro/baby-green, three trays were sampled each at 10, 15, and 20 DAS for nutrient analysis using a pair of scissors to clip seedlings at ground level.

### 2.2. Mature Vegetables

In the 2018 and 2019 growing seasons, tropical spinach (Caribbean Red selection) and roselle (Indian Red-Red) were seeded first week in June and managed conventionally (fertilized with 10-10-10 NPK) under black plastic mulch at Rutgers Hort Farm 3, East Brunswick, NJ and Rutgers Ag Research and Extension Center (RAREC), Bridgeton, NJ, respectively. In 2.2-m (6-foot) plots, tropical spinach seed was sprinkled about 1 cm deep along a slit cut into the plastic mulch at the center of the seedbed. In 3.6-m (10-foot) plots, roselle was seeded about 2 cm deep using 3–4 seed/hole spaced 30 cm (24-inches) apart along the center of the seedbed. Seedlings were thinned to two/hole about 10 DAS. Marketable foliage of tropical spinach was sampled twice at 5 and 7 weeks after sowing (WAS) while marketable foliage from roselle was sampled at 7 and 9 WAS. For tropical spinach, samples were taken from plants within 30 cm at the center of the plot and for roselle, samples were taken from the three plants at the center of the plot.

### 2.3. Plant and Data Analyses

Micro/baby-green and mature vegetable samples with 85–90% moisture were dried to 5–6% moisture in an oven (Wisconsin Oven, Memmert Model, East Troy, WI, USA) set at 50 °C (122 °F) for 72–96 h at College Farm Road on Cook Campus, New Brunswick, NJ. Composite samples of the same treatment taken from three replications were ground using the Thomas Scientific (Swedesboro, NJ, USA) plant sample grinder and mixed. We took two subsamples from each treatment, packaged them carefully and mailed to Brookside Laboratories in New Bremen, Ohio (https://www.blinc.com/ (accessed on 24 September 2021)), for proximate and elemental analyses. The analytical protocols for digestible carbohydrates, digestible protein, dietary fiber, total fat, and macro- and micro-nutrients; and references are shown in Table 1. The data were analyzed using the two subsamples as two replications for each treatment. Means were separated using Tukey’s HSD test at 5% probability level (HSD_05_). The average values of the data obtained for the microgreens at 10, 15, and 20 DAS and the averages obtained from the two sampling dates for the field grown crops (5 and 7 WAS for tropical spinach and 7 and 9 WAS for roselle) were used to compare the partial nutritional status of tropical spinach and roselle micro/baby-greens and mature plants.

## 3. Results

### 3.1. Micro/Baby-Greens and Mature Field Grown Vegetables

Figure 1 shows the stages of growth at which the tropical spinach and roselle micro/baby-greens were sampled for analysis; and approximate growth status of the field grown tropical spinach and roselle at the times of sampling.

### 3.2. Growth Stage Impact on Nutritional Status

#### 3.2.1. Tropical Spinach Micro/Baby-Green

Proximate analysis showed that both digestible carbohydrates and protein changed significantly in the tropical spinach micro/baby-green between 10 and 20 DAS. While the percent digestible carbohydrates doubled from 12.5% to 25% during the 10-day period between 10 and 20 DAS, the digestible protein decreased from 35% to 27% over the same period. The total fat and fiber percent did not change during this period (Figure 2). Elemental analysis showed that percent macronutrient content did not change except for K which dropped from approximately 7% to 6% during this 10-day period. However, the micronutrients Fe and Mn increased between 10 and 20 DAS in the tropical spinach micro/baby-green. The Fe level rose from 98 ppm to 175 ppm while Mn rose from 200 to nearly 300 ppm between 10 and 20 DAS. Copper remained very low (<2%) in the plant during this period while Zn, which was quite high (150–180 ppm), did not change significantly over the same period (Figure 2).

#### 3.2.2. Mature (Field Grown) Tropical Spinach

The digestible carbohydrates and protein were about 25% in mature foliage of tropical spinach at 5 to 7 WAS. Fat was extremely low (<1%) and fiber ranged from 10–15%. In all cases, the growth stage did not impact the nutritional content of tropical spinach foliage significantly (Figure 3). Among the macronutrients analyzed, Ca (2.5–3%) and K (3.5–4.2%) were present at reasonably high levels. Phosphorus and Mg were present at <2% in the mature foliage. The stage of growth showed an impact only on Ca content which was significantly higher at 7 WAS than at 5 WAS. Among the micronutrients Fe content at 100–118 ppm was highest followed by Mn at 34–77 ppm. Copper was <5 ppm, while Zn was approximately 41 ppm in the mature foliage. Only Mn showed a higher level at 7 WAS. The other elements showed no difference in percent content at 5 and 7 WAS (Figure 3).

#### 3.2.3. Roselle Micro/Baby-Green

As observed in tropical spinach micro/baby-green, proximate analysis showed that digestible carb and protein were present at high levels in roselle micro/baby-green. Digestible carbohydrates increased from 21–30% between 10 and 20 DAS while digestible protein decreased from 35–22% over the same period, Fat was low (<2.5%) and fiber was quite high and increased significantly from 13% at 10 DAS to 18.5% 20 DAS (Figure 4). Potassium at 3.75–4.20% was the predominant macronutrient in roselle micro/baby-green from 10–20 DAS. Phosphorus, Ca and Mg were present at low levels (<2%) from 10–20 DAS. No macronutrient showed significant difference in percent content from 10–20 DAS (Figure 4). Iron level at 220 ppm was highest in roselle micro/baby-green at 10 DAS. The level dropped rapidly to <100 ppm at 15 and 20 DAS. Manganese was high (148–180 ppm) in roselle micro/baby-green and unaffected by growth stage from 10–20 DAS. Copper was very low (<2 ppm) regardless of micro/baby-green growth stage, but Zn was high at 77–98 ppm and unaffected by the growth stage of the micro/baby-green roselle (Figure 4).

#### 3.2.4. Mature (Field Grown) Roselle

Digestible carbohydrate at 36–38% and digestible protein at 18–22% were high in mature roselle foliage at 7 and 9 WAS and so was fiber at 14–17%, but fat was low (<2%). Proximate analysis showed no significant difference in nutritional content between 7 and 9 WAS (Figure 5). Macronutrients were generally low in mature roselle foliage at 7 and 9 WAS. Calcium at 1.6–2.0% and K at 1.3–1.7% were the highest macronutrients found in mature roselle foliage. Phosphorus and Mg were present at <1%. Calcium content in roselle foliage increased with plant age whereas K level decreased. Age of foliage did not affect P and Mg content in roselle (Figure 5). Among the micronutrients, Mn at 86–173 ppm was the most prominent in mature roselle foliage at 7 and 9 WAS followed by Fe at approximately 80 ppm. At 7 WAS, copper at 61 ppm was very high but dropped to 7 ppm at 9 WAS. Zinc content in roselle at approximately 20 ppm was similar at 7 and 9 WAS (Figure 5).

### 3.3. Tropical Spinach Micro/Baby-Green and Mature Plant

#### 3.3.1. Carbohydrates, Protein, Fat, and Fiber

In tropical spinach, the micro/baby-green contained higher digestible protein (32.5%) than the marketable foliage from the field-grown crop (25%). Fat was also higher in the microgreen (2%) than in mature vegetable (<0.5%). However, the digestible carbohydrate was higher in the mature crop (27%) than in the microgreen (17.5%). The dietary fiber content (13%) was the same in the microgreen and mature plant (Figure 6).

#### 3.3.2. Macronutrients

Among the macronutrients, the microgreen contained higher level of potassium (K) (6.7%) than the marketable foliage of mature tropical spinach (4.1%), but Ca was higher in the mature plant (2.9%) than in the microgreen (0.9%). The levels of P (about 1%) and Mg (about 1.5%) were not affected by the growth stage of tropical spinach (Figure 6).

#### 3.3.3. Micronutrients

Tropical spinach microgreen contained much higher levels of Mn (>260 ppm) and Zn (170 ppm) than the mature marketable foliage which contained <50 ppm Mn and <40 ppm Zn. Iron was reasonably high (about 120 ppm) in the micro/baby-green and mature tropical spinach but no statistical difference was observed between the two. Copper was very low (0.5–1 ppm) both in the microgreen and mature tropical spinach (Figure 6).

### 3.4. Roselle Micro/Baby-Green and Mature Plant

#### 3.4.1. Carbohydrates, Protein, Fat, and Fiber

As observed in tropical spinach, digestible protein was higher in roselle microgreen (30%) than in the mature plant (22%). In addition, similar to tropical spinach, digestible carbohydrates were higher in the mature roselle (36%) than in the micro/baby-green (24%). The fiber content of roselle micro/baby-green and the mature foliage was close to 15% and no difference was observed in the two. The total fat content was low (1–2%) and no difference was observed between the micro/baby-green and the mature plant (Figure 7).

#### 3.4.2. Macronutrients

All macronutrients except Ca were present at higher levels in the micro/baby-green than in mature roselle foliage. The most significant were P and K where the elements were two times higher (or more) in the micro/baby-green than in the mature foliage. As observed in tropical spinach, Ca was higher in the mature roselle (1.7%) than in the micro/baby-green (0.9%) (Figure 7).

#### 3.4.3. Micronutrients

Manganese, Fe, and Zn were present in high amounts in roselle micro/baby-green and mature plant, but much higher in the micro/baby-green. Copper was relatively low in both the micro/baby-green and mature plant (1–4 ppm) (Figure 7).

### 3.5. Nutritional Value of Micro/Baby Green and Mature Field Grown Tropical Spinach and Roselle

Table 2 and Table 3 show the nutritional value of micro/baby green and mature field grown foliage of tropical spinach and roselle, respectively, for the macronutrients: carbohydrates, protein, and fat. For the tropical spinach, daily consumption of 100 g (dry weight basis) of the micro/baby green should meet 6.2% carbohydrate, 29.8% protein, and 3.2% fat daily intake value and the mature field grown foliage 9.8% carbohydrates, 22.2% protein, and 0.8% fat daily intake value. For roselle, daily consumption of 100 g (dry weight basis) of the micro/baby green should meet 8.7% carbohydrates, 26.7% protein, and 3.2% fat daily intake value and the mature field grown foliage 13.5% carbohydrates, 20% protein, and 1.6% fat daily intake value. In both plants, micro/baby greens are superior sources of protein compared to the mature field grown foliage. Consuming 100 g (dry weight basis) daily will meet close to 30% of the daily protein intake value. However, they are relatively poor sources of carbohydrates and fat and supply less than 10% of the daily intake values of these nutrients.

## 4. Discussion

### 4.1. Growth Stage Impact on Nutritional Status

Proximate and elemental analyses showed that the growth stage of the micro/baby-green has a significant influence on nutritional content. In tropical spinach and roselle micro/baby-greens, digestible carbohydrates doubled between 10 and 20 DAS, while digestible protein dropped significantly over this period. Fat and fiber did not change significantly. Macronutrients were not affected by growth stage except K, which declined during 10–20 DAS in tropical spinach. Iron and Mn were the micronutrients most responsive to growth stage in micro/baby-greens. In tropical spinach micro/baby-green, Fe and Mn increased significantly between 10 and 20 DAS while Cu and Zn levels were relatively unchanged. In roselle, Fe at 10 DAS was high but level dropped sharply afterwards. The level of the other micronutrients did not change between 10 and 20 DAS. These results suggest that the nutritional benefits derivable from tropical spinach and roselle micro/baby-greens are dependent on growth stage.

In the field grown mature plants, nutritional content (digestible carbohydrates, digestible protein, fat, and fiber) did not change significantly between 5 and 7 WAS for tropical spinach and 7 and 9 WAS for roselle. Among the macronutrients, Ca level was higher and K level lower in the older foliage. Among the micronutrients Mn level was higher in the older foliage both in tropical spinach and roselle. Copper level was higher in the younger foliage at 7 WAS than at 9 WAS in roselle. The levels of Fe and Zn did not change with age of foliage in tropical spinach and roselle. It seemed the dynamics of nutritional changes in tropical spinach and roselle is more pronounced at the micro/baby-green stage than at the mature leaf stage. For nutritional benefits, attention to growth stage would be more critical at the micro/baby-green stage.

### 4.2. Micro/Baby-Green and Mature Plant

In our studies, we found that tropical spinach and roselle micro/baby-greens contain high amounts of digestible protein, macronutrients especially K in tropical spinach and P, K, and Mg in roselle. The micronutrients Mn and Zn are high in both tropical spinach and roselle micro/baby-greens; and roselle is a good source of Fe, one of the most deficient elements in human body globally [17]. However, the mature plants contain higher digestible carbohydrates and Ca. Calcium and K levels in mature plants are of particular interest where bone development and/or health is a challenge. Both tropical spinach and roselle contain substantial amount of dietary fiber (>10%), but this component did not vary with the plant’s growth stage. Tropical spinach micro/baby-green showed higher affinity for total fat than the mature plant. These results showed that micro/baby-greens have the capacity to enhance human nutrition, especially where protein and essential macro and micronutrients may be deficient in traditional diets. They confirm the findings of several researchers that microgreens are a powerhouse of high nutrition for human use [4,11,12,18]. Nursing mothers, children and people in less privileged communities where access to good quality food is limited, should benefit from these nutrient sources. Ebert et al. [4] reported similar protein levels in microgreen and fully grown plants across four cultivars of amaranth (called ‘tropical spinach’ in this paper) in Taiwan. However, our study showed a significant difference in protein content between the micro/baby green and the mature field grown foliage, with the latter containing less protein than the microgreen. Factors such as growth environment (light, soil, moisture regime etc.), plant cultivar, and crop management practices might influence these results.

We project an increase in demand for microgreens in New Jersey and the Mid-Atlantic United States as the community becomes more aware of the nutritional and health values hidden in this powerful repository of essential nutrients and health principles. Work is in progress in North America and globally to determine the impact of light on nutritional quality of microgreens [2,14]. One expects that it should be possible to customize microgreens soon to manufacture unique foods and medicines to address some human nutrition and health issues as we move forward in the 21st Century, as projected in the review by Michell et al. [12].

## 5. Conclusions

Based on dry weight analyses, micro/baby greens of tropical spinach and roselle contain higher levels of digestible protein, P, K, Mg, Fe, Mn, and Zn but lower levels of digestible carbohydrates and Ca compared to field grown mature foliage. Foliage tissues of both species as micro/baby greens are relatively low in total fat, Ca, and Cu. Between 10 and 20 days after sowing (DAS), percent digestible carbohydrates in fresh foliar tissue increased 100% in tropical spinach and 50% in roselle, while digestible protein dropped 21% in tropical spinach and 50% in roselle.

Producing micro/baby-greens is not complicated and may be adopted under relatively unsophisticated circumstances, provided there is adequate light, water, nutrient supply, dependable seed, and good sources for good quality growth media. Micro/baby-greens are not as susceptible to food safety issues as sprouts [7,19], making them more adoptable by the general population. Depending on the plant under consideration, the growth cycle may be as short as 10 days, or less (for wheatgrass as we found in other studies) or a little longer (for roselle).

## Figures and Tables

**Figure 1 foods-10-02546-f001:**
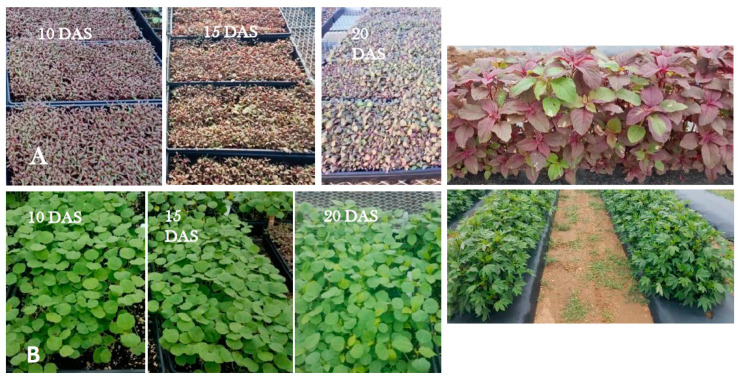
Micro/baby-greens of (**A**) tropical spinach, and (**B**) roselle at 10, 15, and 20 days after sowing (DAS); photos to the right show field grown tropical spinach (Caribbean red) (Top) and roselle (Indian Red-Red) (bottom) at the time of sampling for nutritional status determination. Photos by Albert Ayeni.

**Figure 2 foods-10-02546-f002:**
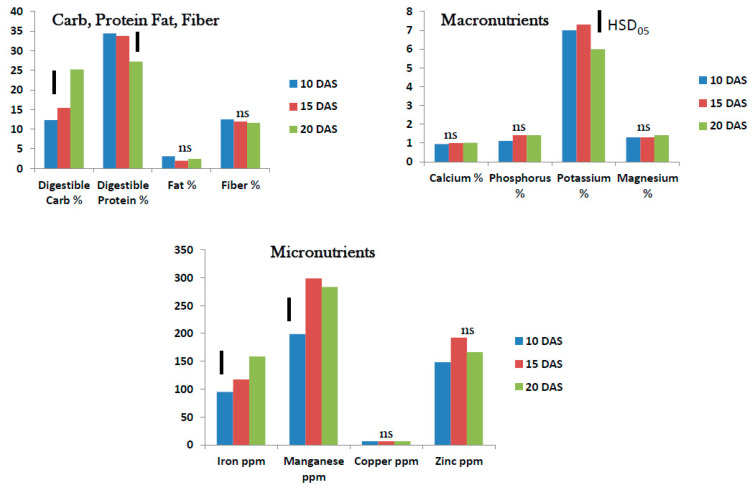
Tropical spinach micro/baby-green: nutrient status (dry weight basis) 10–20 days after seeding (DAS) (ns = no significant difference; bars show Tukey’s HSD at α = 0.05).

**Figure 3 foods-10-02546-f003:**
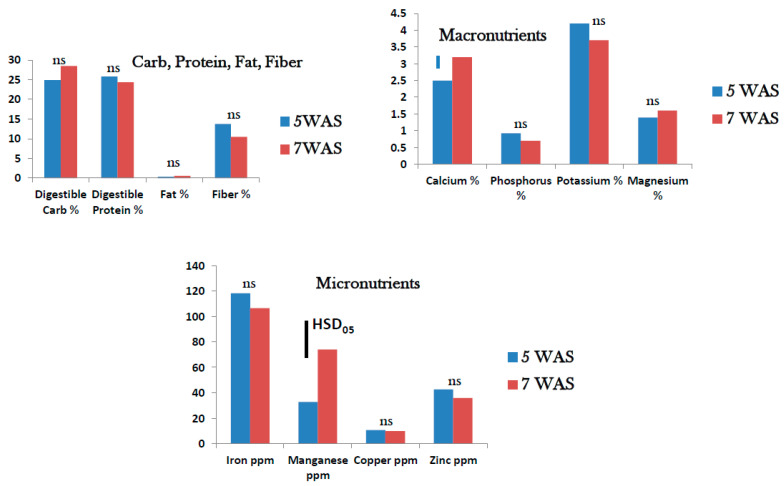
Field grown tropical spinach: Nutrient status 5 and 7 weeks after sowing (WAS) (ns = no significant difference; bars Tukey’s HSD at α = 0.05).

**Figure 4 foods-10-02546-f004:**
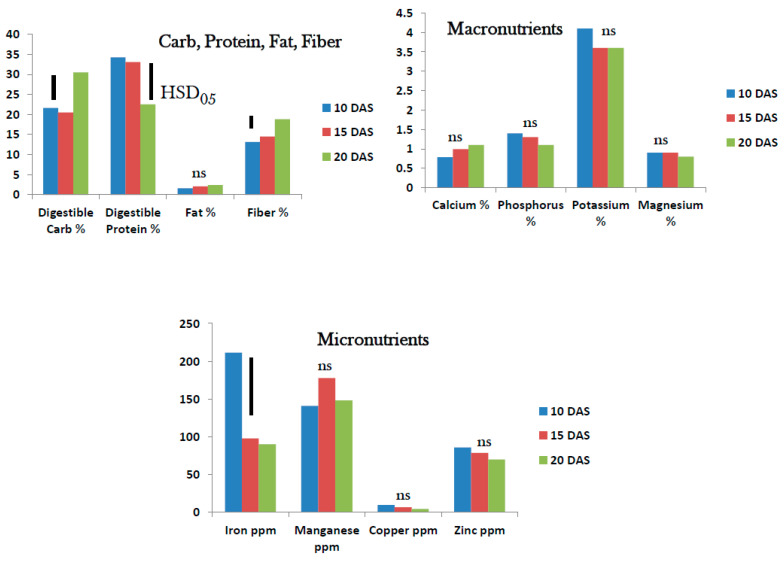
Roselle micro/baby-green: Nutrient status 10–20 days after seeding (DAS) (ns = no significant difference; bars show Tukey’s HSD at α = 0.05).

**Figure 5 foods-10-02546-f005:**
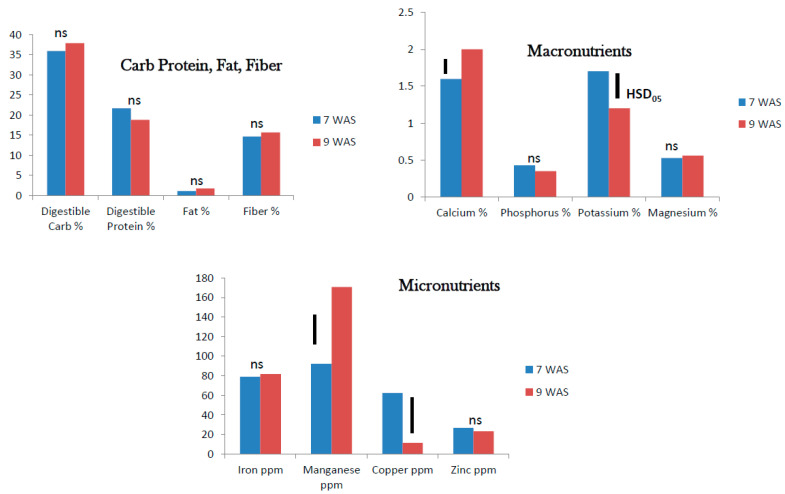
Field grown roselle: Nutrient status 7 and 9 weeks after sowing (WAS) (ns = no significant difference, bars show Tukey’s HSD at α = 0.05).

**Figure 6 foods-10-02546-f006:**
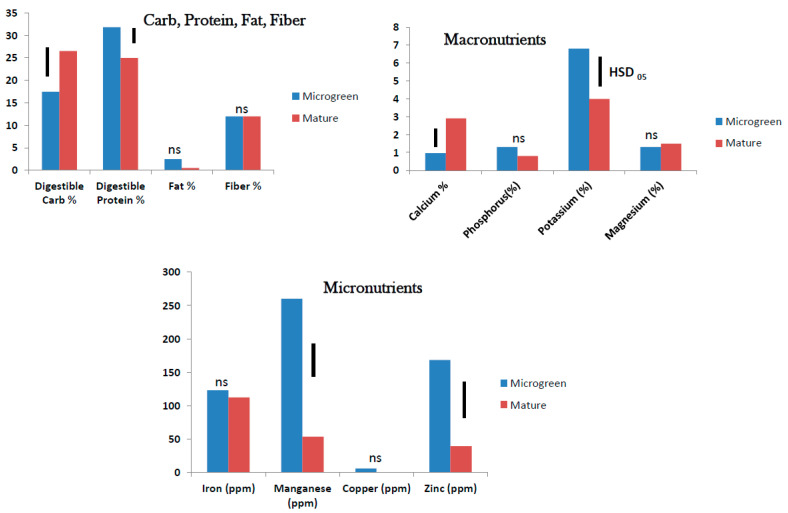
Nutrient status compared on dry weight basis between tropical spinach micro/baby-green (average of 10–20-day old seedlings) and mature plant (average of 5- and 7-week-old plants) (ns = no significant difference; bars show Tukey’s HSD at α = 0.05).

**Figure 7 foods-10-02546-f007:**
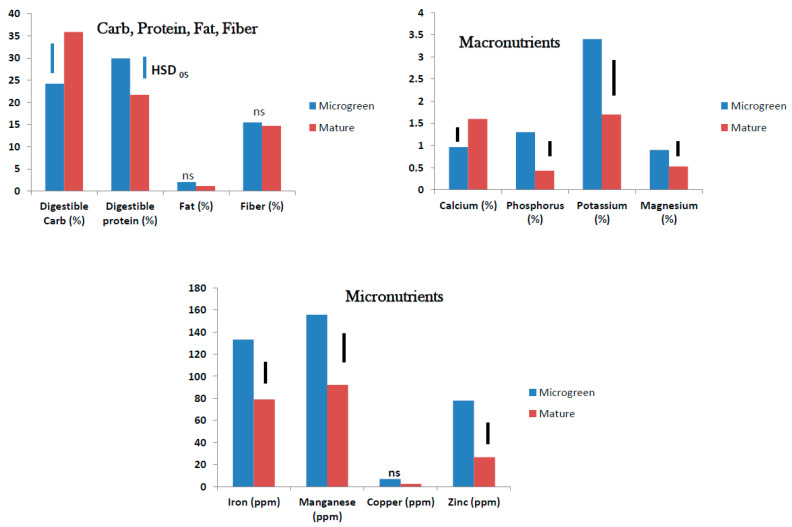
Nutrient status compared on dry weight basis between roselle micro/baby-green (average of 10–20 day-old seedlings) and mature plant (average of 7- and 9-week-old plants). (ns = no significant difference; bars show Tukey’s HSD at α = 0.05).

**Table 1 foods-10-02546-t001:** Plant analytical methods for digestible carbohydrates, digestible protein, dietary fiber, total fat, and elemental macro-and micro-nutrients (based on dry matter samples).

Plant Nutrient	Analytical Protocol	Reference
Digestible Protein	AOAC 990.03 Combustion analysis by Elementar Max Exceed	[13]
Dietary Fiber	AOAC 962.09 crude fiber by filter method	[14]
Total Fat	AOAC 920.39 crude fat by ether extract	[15]
Macro and Micro Elements	Modified AOAC 985.01, metals by ICP determination (digestion by nitric acidin Teflon vessels digested in a CEM Mars Express microwave)	[16]
Digestible Carbohydrates	Calculated	[15]

**Table 2 foods-10-02546-t002:** Tropical spinach: nutritional value of 100 g (dry weight) micro/baby green and mature field grown foliage. Figures show average for 10-, 15-, and 20-day-old micro/baby greens and average for 5- and 7-week-old field grown plants. Daily value assumptions: Carbohydrates—1100 calories (Range 900–1300); Protein—450 calories (Range 200–700); Fat—550 calories (Range 400–700). Calorie estimation: 1 g Carbohydrates or Protein = 4 calories; 1 g Fat = 9 calories.

Nutritional Item	Micro/Baby Green	% Daily Value	Field Grown Mature Foliage	% Daily Value
Digestible Carb (%)	17 (68 calories/100 g)	6.2	27 (108 calories/100 g)	9.8
Digestible Protein (%)	33.5 (134 calories/100 g)	29.8	25 (100 calories/100 g)	22.2
Total Fat (%)	2 (18 calories/100 g)	3.2	0.5 (4.5 calories/100 g)	0.8

**Table 3 foods-10-02546-t003:** Roselle: nutritional value of 100 g (dry weight) micro/baby green and mature field grown foliage. Figures show average for 10-, 15-, and 20-day-old micro/baby green and average for 7- and 9-week-old field grown plants. Daily value assumptions: Carbohydrates—1100 calories (Range 900–1300); Protein—450 calories (Range 200–700); Fat—550 calories (range 400–700). Calorie estimation: 1 g Carbohydrates or Protein = 4 calories; 1 g Fat = 9 calories.

Nutritional Item	Micro/Baby Green	% Daily Value	Field Grown Mature Foliage	% Daily Value
Digestible Carb (%)	24 (96 calories/100 g)	8.7	37 (148 calories/100 g)	13.5
Digestible Protein (%)	30 (120 calories/100 g)	26.7	22.5 (90 calories/100 g)	20
Total Fat (%)	2 (18 calories/100 g)	3.2	1 (9 calories/100 g)	1.6

## Data Availability

This paper did not report any data external to the study.

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
