# Peer review of "Nutrient Content of Micro/Baby-Green and Field-Grown Mature Foliage of Tropical Spinach (Amaranthus sp.) and Roselle (Hibiscus sabdariffa L.)"

_foods, 2021, doi:10.3390/foods10112546_

Round 1

Reviewer 1 Report

The article entitled “Nutrient content of micro/baby-green and field-grown mature foliage of tropical spinach (Amaranthus sp.) and roselle (Hibiscus sabdariffa L.)” provides the data on the composition of digestible carbohydrates, digestible protein, dietary fiber, and minerals in micro/baby-green and field-grown mature foliage of tropical spinach (Amaranthus sp.) and roselle (Hibiscus sabdariffa L.).

I have the following comments for this article:

Abstract: The sentence “Fresh foliar products from both vegetables have high relative percentages of P, K, and Mg; and relatively high fresh weight ppm Fe, Mn, and Zn” is unclear. Are the data mentioned in the article on are fresh weight basis or dry weight basis? Please specify.

Material and methods: “Micro/baby-green and mature vegetable samples were dried in an oven set at 50 oC (122 oF) for 72-96 hours” what are the moisture contents in the fresh and dehydrated vegetables?.

The methods of analysis of digestible carbohydrates, digestible protein, dietary fiber, and minerals are missing.

Results: In the headings of Figure 2-7, please specify the contents on a fresh weight basis or dry weight basis.

Figure 6 and 7: in these figures, the composition of micro/baby-green are compared with field-grown mature foliage. Here, please specify the age of plants, for instance, 10, 15, or 20 DAS for micro/baby-green and 7 or 9 WAS for field-grown mature foliage.

Discussion: need a significant improvement. Please provide a discussion on the contents of these elements recorded in previous studies on these micro/baby-green and field-grown mature foliage of tropical spinach (Amaranthus sp.) and roselle (Hibiscus sabdariffa L.)? your data are following previous studies, or there is a significant variation? Please discuss.

Considering the daily intake of micro/baby-green and field-grown mature foliage and nutrient contents recorded in this study, please calculate and discuss the % Daily Value of these nutrients. For instance, intake of 100g tropical spinach micro/baby-green can provide X% of the daily value.

Conclusion: Need revision, please provide the summary of your data. The sentences between lines 294-301 can be shifted to the discussion section. 

Author Response

Reviewer 1

Comments and Suggestions for Authors

The article entitled “Nutrient content of micro/baby-green and field-grown mature foliage of tropical spinach (Amaranthus sp.) and roselle (Hibiscus sabdariffa L.)” provides the data on the composition of digestible carbohydrates, digestible protein, dietary fiber, and minerals in micro/baby-green and field-grown mature foliage of tropical spinach (Amaranthus sp.) and roselle (Hibiscus sabdariffa L.).

I have the following comments for this article:

Abstract: The sentence “Fresh foliar products from both vegetables have high relative percentages of P, K, and Mg; and relatively high fresh weight ppm Fe, Mn, and Zn” is unclear. Are the data mentioned in the article on are fresh weight basis or dry weight basis? Please specify.

Both vegetables have high relative percentages of P, K, and Mg; and relatively high ppm Fe, Mn, and Zn.. The data in the article are on dry weight basis. This has been specified in the revised manuscript

Material and methods: “Micro/baby-green and mature vegetable samples were dried in an oven set at 50 oC (122 oF) for 72-96 hours” what are the moisture contents in the fresh and dehydrated vegetables?.

Fresh Micro/baby-green vegetable samples contained 85-90% moisture; dehydrated contains 5-6% moisture

The methods of analysis of digestible carbohydrates, digestible protein, dietary fiber, and minerals are missing. See analytical methods provided and referenced in Table 1

Results: In the headings of Figure 2-7, please specify the contents on a fresh weight basis or dry weight basis. Done

Figure 6 and 7: in these figures, the composition of micro/baby-green are compared with field-grown mature foliage. Here, please specify the age of plants, for instance, 10, 15, or 20 DAS for micro/baby-green and 7 or 9 WAS for field-grown mature foliage. Done

Discussion: need a significant improvement. Please provide a discussion on the contents of these elements recorded in previous studies on these micro/baby-green and field-grown mature foliage of tropical spinach (Amaranthus sp.) and roselle (Hibiscus sabdariffa L.)? your data are following previous studies, or there is a significant variation? Please discuss. I cited the work of Ebert et al (2015) where they reported similar levels of protein in the microgreen and fully grown amaranths. This is a variation from our observation. They expressed concern about this observation as they expected to see a difference between the sprout, the microgreen and the fully grown plants compared in their study. There is literature on light, soil, and other environmental issues that might influence the nutrient status of these plants which may vary significantly from one location to another..

Considering the daily intake of micro/baby-green and field-grown mature foliage and nutrient contents recorded in this study, please calculate and discuss the % Daily Value of these nutrients. For instance, intake of 100g tropical spinach micro/baby-green can provide X% of the daily value. Section 3.5  and Tables 2 and 3 have been added to address this comment

Conclusion: Need revision, please provide the summary of your data. The sentences between lines 294-301 can be shifted to the discussion section Done.

Reviewer 2 Report

Brief summary.

Interesting study that provides comparative basic information on the nutritional content of tropical spinach and the roselle, including digestible carbohydrates, digestible protein, total fat, dietary fibber, and elemental macro-and micro-nutrients as both greenhouse-cultured micro/baby-green and mature field grown vegetable stages.

Broad comments.

Introduction should conclude with an explicit working hypothesis and concrete objectives.

Methods of analysis used for carbohydrate, protein, fat and fibre determinations, as well as for determining macro and micronutrient content, should be referenced, or described, on. 2.3.

There is an incongruity between the statistical methods used to separating means: Tukey's test (Apt. 2.3) or LSD (Figs 1 to 7).

Data could be presented in table format, rather than figures, to reduce space and facilitate general interpretation of the behaviour of both micro-greens and mature plants and would facilitate the comparison between both development levels. In this way, Figures 6 and 7 would be unnecessary.

Poor discussion and limited to repeating data behaviour already described on Results, should be improved using updated references.

Author Response

Reviewer 2

Comments and Suggestions for Authors

Brief summary.

Interesting study that provides comparative basic information on the nutritional content of tropical spinach and the roselle, including digestible carbohydrates, digestible protein, total fat, dietary fiber, and elemental macro-and micro-nutrients as both greenhouse-cultured micro/baby-green and mature field grown vegetable stages.

Broad comments.

Introduction should conclude with an explicit working hypothesis and concrete objectives. Done

Methods of analysis used for carbohydrate, protein, fat and fibre determinations, as well as for determining macro and micronutrient content, should be referenced, or described, on. 2.3.

Table 1 has been added to clarify analytical methods

There is an incongruity between the statistical methods used to separating means: Tukey's test (Apt. 2.3) or LSD (Figs 1 to 7). This error is regretted. The figures with LSD were sent in error. The correct figures with HSD and appropriate captions are attached

Data could be presented in table format, rather than figures, to reduce space and facilitate general interpretation of the behaviour of both micro-greens and mature plants and would facilitate the comparison between both development levels. In this way, Figures 6 and 7 would be unnecessary. I prefer to keep the data as figures for Figures 6 and 7. I read figures more conveniently than Tables in this type of data presentation.

Poor discussion and limited to repeating data behaviour already described on Results, should be improved using updated references.  Discussion and Conclusions have been revised. Two major references have been added to update the literature review and citation.

Round 2

Reviewer 1 Report

NA